# Protocol for the economic evaluation of COVID-19 pandemic response policies

Brandon Wen Bing Chua ![ORCID],[1] Vinh Anh Huynh,[1] Jing Lou,[1] Fang Ting Goh,[1] Hannah Clapham ![ORCID],[1] Yot Teerawattananon,[1,2] Hwee Lin Wee[1,3]

[1]Saw Swee Hock School of Public Health, National University of Singapore, Singapore
[2]Health Intervention and Technology Assessment Program (HITAP), Ministry of Public Health, Nonthaburi, Thailand
[3]Department of Pharmacy, National University of Singapore, Singapore

**Correspondence to**
Dr Hwee Lin Wee;
weehweelin@nus.edu.sg

## ABSTRACT

**Introduction** Several treatment options are available for COVID-19 to date. However, the use of a combination of non-pharmaceutical interventions (NPIs) is necessary for jurisdictions to contain its spread. Although the implementation cost of NPIs may be low from the healthcare system perspective, it can be costly when considering the indirect costs from the societal perspective. COVID-19 vaccination campaigns have begun in several countries worldwide. Nonetheless, the quantity of vaccines available remain limited over the next 1 to 2 years. A tool for informing vaccine prioritisation that considers both cost and effectiveness will be highly useful. This study aims to identify the most cost-effective combination of COVID-19 response policies, using Singapore as an example.

**Methods and analysis** An age-stratified Susceptible-Exposed-Infectious-Recovered model will be used to generate the number of infections stratified by disease severity under different intervention scenarios. Polices of interest include test-trace-isolate, travel restriction, compulsory face mask and hygiene practices, social distancing, dexamethasone/remdesivir therapy and vaccination. The latest phase 3 trial results and the WHO Target Product Profiles for COVID-19 vaccines will be used to model vaccine characteristics. A cost (expected resource utilisation and productivity losses) and quality-adjusted life years (QALYs) will be attached to these outputs for a cost-utility analysis. The primary outcome measure will be the incremental cost-effectiveness ratio generated from the incremental cost of policy alternatives expressed as a ratio of the incremental benefits (QALYs gained). Efficacy of policy options will be gathered from literature review and from its observed impacts in Singapore. Cost data will be gathered from healthcare institutions, Ministry of Health and published data. Sensitivity analysis such as threshold analysis and scenario analysis will be conducted.

**Ethics and dissemination** Ethics approval was not required for this study. The study findings will be disseminated through peer-reviewed journals.

## INTRODUCTION

The coronavirus disease 2019 (COVID-19) is a highly transmissible infectious disease caused by a novel severe acute respiratory syndrome coronavirus 2 (SARS-CoV-2). As of 16 March 2021, COVID-19 has affected more than 120 million people and resulted in over 2.6 million deaths worldwide.[1]

### Strengths and limitations of this study

► Robust age-stratified Susceptible-Exposed-Infectious-Recovered model with an extensive contact matrix of 16 age groups, 2 subpopulations and 3 different locations of interest.
► Various combination of non-pharmaceutical interventions and pharmaceutical drugs are analysed with vaccine distribution strategies.
► The proposed costing framework may be adapted for various jurisdictions to guide timely collection of essential variables to accelerate decision-making process.

According to the Asian Development Bank, up to US$8.8 trillion in global economic loss is expected due to COVID-19, equivalent to approximately 10% of the world gross domestic product (GDP).[2]

We have now entered a very exciting time with at least three COVID-19 vaccines gaining market authorisation. Vaccination has already begun in several countries such as the USA, UK, Russia, China, Israel and United Arab Emirates.[3] Nonetheless, the quantity of vaccines available remain limited over the next 1 to 2 years. A tool for informing vaccine prioritisation that takes cost and effectiveness into account will be highly useful.

Mathematical and computational simulation models can support policymakers in decision-making. A total of seven different models have been evaluated by the COVID-19 Multi-Model Comparison Collaboration, all of which describe disease transmission mechanistically to capture the effect of interventions on transmission and predict the impact of interventions.[4] The models project the number of infections by different severity, allowing for a variety of health-related and resource use results to be evaluated. By attaching costs and health utilities to the output of these models, we will be able to evaluate the cost-effectiveness of various policy scenarios. Given the scarcity of both healthcare and non-healthcare resources,

techniques such as cost-effectiveness analysis provides a means of efficiently allocating scarce resources.

To facilitate efficient resource allocation, this study protocol aimed to describe a framework for conducting economic evaluation of pandemic response policies in the context of COVID-19. A new framework is required given that non-pharmaceutical interventions (NPIs) play a much more important role compared with other viral infections such as influenza virus or varicella-zoster virus. The earlier cost-effectiveness models for influenza vaccine or varicella-zoster vaccine did not consider NPIs in an elaborate manner. Many challenges exist with conducting an economic evaluation involving COVID-19 response strategies especially with the dearth of vaccine data. At the same time, such evaluations may not be feasible on the release of vaccine candidates, as prompt decisions by policymakers may be required within a limited time frame. Hence, this paper will also serve to guide the timely collection of key data fields before the arrival of vaccines, especially in settings where expertise in economic evaluation is limited.

## METHODS AND ANALYSIS

This paper describes the protocol of an economic evaluation of COVID-19 response measures with examples from the Singapore healthcare sector and societal perspective. The healthcare sector perspective, recommended by the Agency for Care Effectiveness (ACE) in Singapore,[5] is necessary to evaluate the impact of managing COVID-19, especially in circumstances where there are capacity limitations. The societal perspective, although not recommended by ACE, has been recommended by the Second Panel on Cost-Effectiveness in Health and Medicine,[6] and is essential for understanding the broader social and economic costs attributed to the management of COVID-19. The methods and reporting of the study results will conform to the Consolidated Health Economic Evaluation Reporting Standards.[7] The study will commence on 01 October 2021 with a proposed completion date of 30 June 2022.

## Cost-effectiveness analysis

A cost-effectiveness analysis (CEA) provides a framework for comparing the incremental cost and incremental benefits of alternative policy options. To do so, the costs and benefits associated with each policy option must be estimated separately. A Markov cohort model is commonly used in CEA to simulate data beyond the observed period and also to integrate data inputs from multiple sources.[8] However, a Markov cohort model will be less appropriate for an infectious disease where considerations of dynamic transmission is critical to ensure an accurate prediction of events. The indirect effects of averted infections are also not captured in Markov models, potentially undervaluing the effects of an intervention.[9] In the context of a vaccination programme, this is significant as the risk of infection among both vaccinated and non-vaccinated susceptible people decreases with vaccination as the extent of virus circulating in the community is reduced.

Hence, in this paper, we propose to adopt a Susceptible-Exposed-Infectious-Recovered (SEIR) transmission model (to be elaborated in the next section) and to attach cost and quality-adjusted life years (QALYs) to the outputs from the SEIR model (number of infections by disease severity). The difference in costs between the reference case and the policy alternative is divided by the difference in QALYs between the reference case and the policy alternative to obtain an incremental cost effectiveness ratio (ICER). The ICER will then be interpreted according to the willingness-to-pay threshold of the jurisdiction of interest.

In figure 1, we provide an illustration of the possible combinations of policy options that should be evaluated.

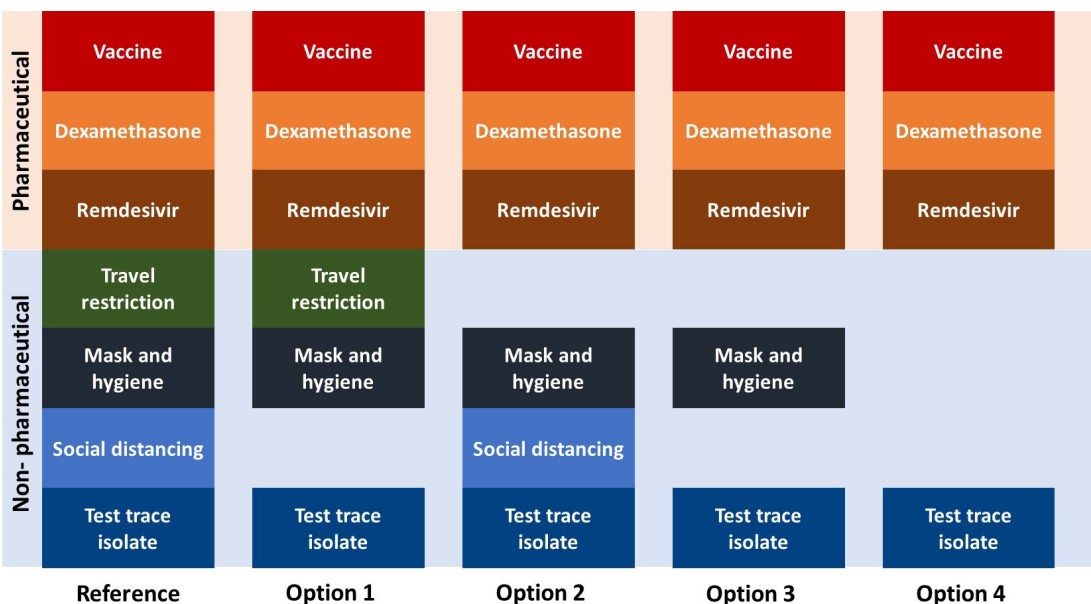

**Figure 1** Combination of COVID-19 pandemic response policies.

This list is derived from discussions with infectious disease clinicians, policymakers, health technology assessment experts and COVID-19 disease modellers. As an example in the Singapore context, the use of test-trace-isolate (TTI), mask and hygiene and treatments for COVID-19,[10] are central to all policy strategies as they are not expected to be removed in the presence of a COVID-19 vaccine, particularly when vaccine supply is limited and 100% efficacy of the vaccine cannot be assumed. However, this should be adapted to each jurisdiction's context. In the reference case, vaccination, travel restriction and social distancing are considered together with the three policies described above. In the presence of the vaccine, the extent of relaxation of policies can be explored. This can be the complete removal of travel restriction (option 2) or social distancing (option 1) or both (option 3). In addition, a partial relaxation of all policy options except TTI in the reference case can be explored in the presence of a vaccine (option 4).

The key vaccine characteristics to be considered in this study will be guided by the WHO Target Product Profile (TPP),[11] as well as emerging data on the Pfizer, Moderna and Oxford-AstraZeneca vaccines.[12–14] This include the indication of use, contradiction, target population, efficacy, safety, contraindications, dose regime, duration of protection, route of administration and product stability.

The vaccination distribution strategies to be considered is determined by vaccine availability and population size to be vaccinated. Given the global demand for COVID-19 vaccines, it is unlikely that sufficient supplies are available for an entire country for several months from the release of a commercially available vaccine. Furthermore, manufacturers may choose to supply vaccines only to those who are willing to pay high prices. Hence, varying levels of vaccine supply availability should be examined in scenario analyses (eg, 100 000 units per month, 500 000 units per month and 1 000 000 units per month). In addition, varying vaccine efficacy and prioritisation of who should receive the limited vaccines will be considered for each vaccine distribution strategy.

## Model description

An age-structured extended SEIR model will be programmed in R using a differential equation model to generate the number of infections in Singapore over time by age group and stratified by disease severity (asymptomatic, mild–moderate, severe, very severe). A schematic outline of the transmission model is depicted in figure 2. The population will be split into two subpopulations: those residing in the foreign worker dormitories (FWD) and those residing outside of FWD. This allows for different COVID-19 policies between the two

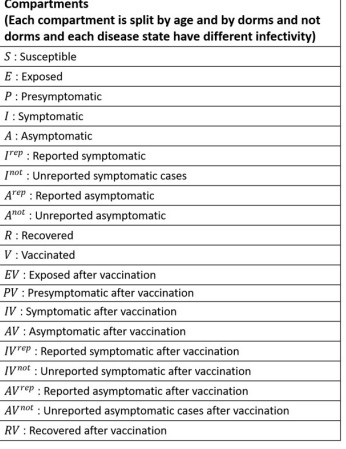

**Compartments**
(Each compartment is split by age and by dorms and not dorms and each disease state have different infectivity)

| | |
|---|---|
| $S$ : Susceptible | |
| $E$ : Exposed | |
| $P$ : Presymptomatic | |
| $I$ : Symptomatic | |
| $A$ : Asymptomatic | |
| $I^{rep}$ : Reported symptomatic | |
| $I^{not}$ : Unreported symptomatic cases | |
| $A^{rep}$ : Reported asymptomatic | |
| $A^{not}$ : Unreported asymptomatic | |
| $R$ : Recovered | |
| $V$ : Vaccinated | |
| $EV$ : Exposed after vaccination | |
| $PV$ : Presymptomatic after vaccination | |
| $IV$ : Symptomatic after vaccination | |
| $AV$ : Asymptomatic after vaccination | |
| $IV^{rep}$ : Reported symptomatic after vaccination | |
| $IV^{not}$ : Unreported symptomatic after vaccination | |
| $AV^{rep}$ : Reported asymptomatic after vaccination | |
| $AV^{not}$ : Unreported asymptomatic cases after vaccination | |
| $RV$ : Recovered after vaccination | |

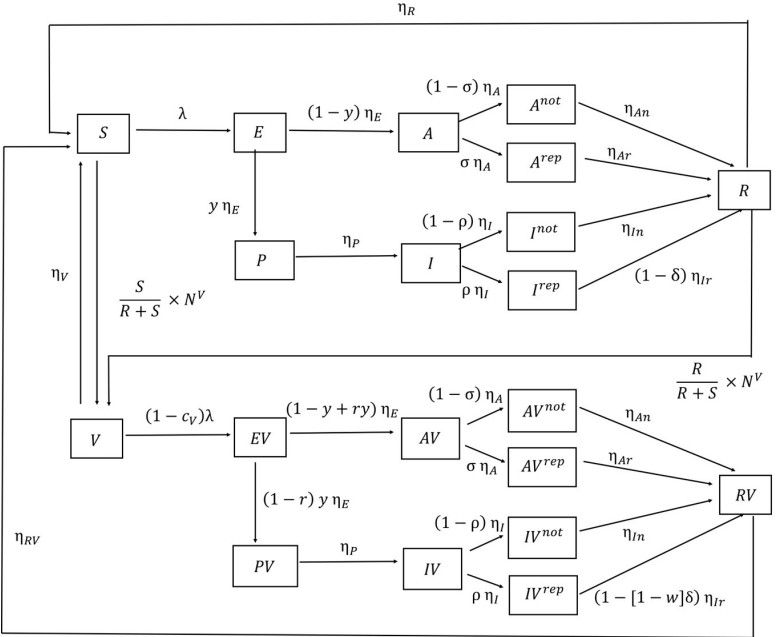

**Parameters (May depend on age and on dorms and not dorms)**

| |
|---|
| $y$ : Probability of developing symptoms after getting infected |
| $\lambda$ : Force of infection (dependent on contact matrices and proportion of infected individuals) |
| $\sigma$ : Proportion of asymptomatic cases that are reported |
| $\rho$ : Proportion of symptomatic cases that are reported |
| $\delta$ : Proportion of deaths among reported symptomatic cases |
| $c_V$ : Effectiveness of vaccine in reducing probability of getting infected from an infectious individual |
| $r$ : Effectiveness of vaccine in reducing probability of being symptomatic after being infected |
| $w$ : Effectiveness of vaccine in reducing probability of death |
| $N^V$: Number of people who got vaccinated per unit time |
| $\eta_E$ : Incubation period |
| $\eta_P$ : Duration of presymptomatic state |
| $\eta_A$ : Time delay in reporting of asymptomatic cases |
| $\eta_I$ : Time delay in reporting of symptomatic cases |
| $\eta_{Ar}$ : Duration of stay in isolation facilities |
| $\eta_{An}$ : Recovery time for unreported asymptomatic cases after time delay in reporting |
| $\eta_{Ir}$ : Duration of stay in healthcare facilities until recovery |
| $\eta_{In}$ : Recovery time for unreported symptomatic cases after time delay in reporting |
| $\eta_V$ : Duration of vaccine protection |
| $\eta_R$ : Duration of immunity acquired from infection |
| $\eta_{RV}$ : Duration of immunity acquired from infection for vaccinated individuals |

**Figure 2** Susceptible-Exposed-Infectious-Recovered transmission model schematics.

subpopulations as implemented in Singapore (eg, more aggressive testing policy in FWD) as well as to capture the very different transmission in the two groups. Each subpopulation is further stratified into 16 age groups of 5 years interval starting from age 0. This is to account for varying probabilities of getting infected and developing symptoms at different severity levels and vaccine effectiveness according to age groups. The effectiveness of vaccines will be modelled based on the following parameters with vaccine reducing the (1) probability of getting infected when in contact with an infectious individual; (2) probability of being symptomatic after being infected; (3) probability of progression to severe disease post infection; and (4) infectiousness of infected individuals.

The interaction between the 32 groups (16 in FWD and 16 outside FWD) is described by contact matrices. For example, a subpopulation with 16 age groups at a location of interest (school/workplace/community) will generate a 16×32 contact matrix. The matrix will describe the average number of contacts with others made by an individual in a location of interest. The overall contact matrix for a subpopulation is a linear combination of the three location-specific contact matrices, with the coefficient for each location-specific matrix ranging between 0 and 1. The implementation of NPIs, which reduces interaction between individuals could be modelled by adjusting the relevant contact matrix coefficient (eg, school closure modelled as a drop in the school contact matrix coefficient).

With the contact matrices and the probabilities of getting infected and developing symptoms, we could estimate the transmission rate for each age group in each subpopulation, and in turn estimate the force of infection at each time. For parameters for which there is great uncertainty or that influence the outcomes greatly, we will conduct sensitivity analyses, and assess how these results impact the conclusions reached. The entire population of Singapore based on population norms in 2019 will be simulated over a period of 1 year.

### Healthcare utilisation

We assume that for the general population there is no active surveillance although there are pockets of individuals such as those in high risk occupations who may need to undergo active surveillance. In addition, we assume that patients with asymptomatic infections do not incur any healthcare resources until they develop symptoms. In Singapore, all symptomatic patients will be isolated within the hospital. Patients with mild-to-moderate symptoms will be managed in isolation wards. Patients with severe symptoms will require ventilator, but not intensive care. Patients with very severe symptoms will require intensive care.

### COVID-19 response policies (pharmaceutical and NPI) and its impacts

The COVID-19 response policies included in this study are detailed in table 1 and their impacts will be simulated in the transmission model. Briefly, the policies include: (1) TTI; (2) imposing travel restriction (international and/or domestic travel); (3) compulsory mask and improved hygiene practices; (4) social distancing: including workplace closures, school closures and work from home policies; (5) dexamethasone and/or remdesivir use in patients with COVID-19 with severe and very severe symptoms; and (6) COVID-19 vaccination. A systematic review will be conducted to obtain efficacy estimates for the policies against COVID-19. The estimated impact of each policy, in the form of reduction in contacts, COVID-19 case numbers or disease severity, will be used for this study. Data from the Singapore context will be used as much as possible, failing which, data sources from developed Asian economies such as Hong Kong, Korea and Japan will be considered. The efficacy and safety data from the latest COVID-19 vaccine trials will be used,[12–14] and supplemented with the WHO TPP.[11] The most relevant control policies at the time of study completion will be reported.

### Measurement and valuation of health benefits

Health benefits will be measured using QALYs. This is derived by multiplying the number of life years lived with a health utility weight to adjust for health-related quality of life (HRQoL). Published health utilities among patients with COVID-19 will be used.[15–17] When data on impact of long COVID-19 on HRQoL emerge,[18] these may also be incorporated into the model. As medical isolation has shown to have negative impacts on HRQoL due to depression and anxiety, we will apply a reduction of QALY on all patients who were isolated for COVID-19.[19] In addition, we will estimate QALY losses due to vaccine-related adverse events based on influenza vaccine for mild/injection site reactions, severe anaphylaxis and severe grade 3 reactions requiring hospitalisation.[20]

### Measurement and valuation of cost

The summary of the sources of cost data and valuations are shown in table 2. All costs will be expressed in 2021 Singapore dollars (SG$) as we expect that a vaccine will arrive only in 2021. This may be tweaked according to local circumstances as some countries have begun mass vaccination in December 2020. The analysis will be done for a time horizon of 1 year, assuming that the efficacy of the vaccine only lasts for 1 year. Hence no discounting will be applied for this study. Alternative time horizons and discount rates may be explored in scenario analyses.

Based on the healthcare system's perspective, direct medical costs will include costs to manage the infections stratified by severity, cost to manage adverse events associated with the vaccine and costs of the vaccination programme. Costs to manage the infections include hospitalisation, medications, laboratory tests, diagnostic tests, radiological examinations, use of ventilators, medical procedures and so on, where applicable. The cost of COVID-19 vaccine will be based on the acquisition price, cost of accompanying delivery device (if any) and

**Table 1** Key COVID-19 response policies and descriptions

| Policies | Subpolicies | Descriptions |
| --- | --- | --- |
| Test-trace-isolate policy | Community testing | A combination of active testing within the community to identify positive cases, establish the chain of transmission and isolate positive cases to prevent further spread. |
| | Isolation of positive cases (or suspected positive cases) | Policy to require mild/moderate positive patients to be isolated either in their homes or in care facilities. This includes policy requiring international travellers to observe self-isolation for a fixed period of time. Where extensive contact tracing is available, people who have come into close contacts with patients who are COVID-19 positive are also required to serve stay-at-home notice or quarantine orders. |
| | Contact tracing initiatives | Government or state's initiatives to identify close contacts of COVID-19 cases to prevent the spread. |
| Travel restriction | – | International travel restriction implemented by the government to ban, limit or restrict international travellers. |
| Compulsory face masks and improved hygiene practices | – | Requirement to wear face masks or facial coverings when in public spaces. |
| Social distancing | As-needed workplaces and schools closures | Requirement to shut down workplaces or educational facilities when a cluster or a number of cases are detected, in order to test exposed individuals and disinfect the affected locations. |
| | Compulsory work-from-home policies | This policy is complementary with social lockdown. Companies are to continue its operation with their staff working from home. This policy may remain in place even after social lockdown has been lifted. |
| | Physical distancing | Sets of policies requiring social gathering to maintain minimum distance (1 to 2 m) between members when outside of their homes or residences. This policy is often accompanied by restricting service capacity of business, mainly retails, malls and personal services to prevent overcrowding. |
| Dexamethasone therapy | – | Intravenous dexamethasone 6 mg one time per day for up to 10 days among patients with severe and very severe symptoms. |
| Remdesivir therapy | – | Intravenous remdesivir 200 mg on day 1, followed by 100 mg one time per day up to day 10 among patients with severe and very severe symptoms. |
| Vaccination | – | Vaccination against SARS-CoV-2. |

the costs associated with cold-chain management when it becomes available but may otherwise be estimated based on reported spending of UK and US government on COVID-19 vaccine deals.[21 22]

From the societal perspective, direct non-medical costs such as childcare services and indirect cost such as lost productivity will be included. There are two possible approaches to quantifying the economic cost of NPIs. One approach, which is being explored by our team, is to use stock market performance as a proxy. This is based on the assumption that the stock market performance is sensitive to any announcement of NPIs, for example, stock market index falls following announcement of border closure due to concerns over the impact on businesses. The change in stock market indices may then be tied to GDP to quantify the economic losses in absolute dollar terms. The reason for not using GDP directly is because GDP data are reported on a quarterly basis and may

not be as sensitive to announcements of NPI compared with stock market indices. This approach of using stock market index as a proxy has been used in other studies to measure investors' sentiment and expected economic activities, which in turn reflect investors' own perceptions of the economic impacts of NPIs.[23 24] Another approach is more labour intensive and is elaborated in the online supplemental appendix.

### Model calibration and validation

Model calibration is necessary to ensure that model predictions are consistent with data sources informing the model inputs.[25] Hence, model parameter estimates will be calibrated with relevant empirical evidence. This can be done using the Bayesian approach, which involves defining probability distributions for model parameters, while considering evidence on the distribution of modelled outcomes and assumptions that relate model

**Table 2** Key study components and recommended sources

| | | Units | Remarks | Sources |
|---|---|---|---|---|
| **Medical cost** | **COVID-19 treatment cost** | | | |
| | Mild/moderate | SG$/case | Estimated | Hospital database, Ministry of Health |
| | Severe | SG$/case | Estimated | Hospital database, Ministry of Health |
| | Very severe | SG$/case | Estimated | Hospital database, Ministry of Health |
| | Death | SG$/case | Estimated | Hospital database, Ministry of Health |
| | **Vaccine-related cost** | | | |
| | Cost to treat adverse side effect (mild) | SG$/case | Estimated | Literature review, COVID-19 vaccine trial data (when available) |
| | Cost to treat adverse side effect (severe) | SG$/case | Estimated | Literature review, COVID-19 vaccine trials (when available) |
| | Cost per dose | SG$/dose | Estimated | Government purchase agreements, hospital and clinical charges |
| | Cost of vaccine delivery device | SG$/dose | Estimated | Government purchase agreements, hospital and clinical charges |
| **Productivity loss** | **Number of productive days loss** | | | |
| | Mild/moderate | Days | Based on length of hospital stay | Hospital database, Ministry of Health |
| | Severe | Days | Based on length of hospital stay | Hospital database, Ministry of Health |
| | Very severe | Days | Based on length of hospital stay | Hospital database, Ministry of Health |
| | Death | Days | Based on length of hospital stay | Hospital database, Ministry of Health |
| | **Population parameters** | | | |
| | Average age of infection/ population | Years | Calculated | Hospital database, Ministry of Health |
| | Age of retirement | Years | Estimated | Ministry of Manpower/Labour |
| | Expected life years | Years | Estimated | Worldometer, WHO |
| | Employed persons | Persons | Estimated | Ministry of Manpower/Labour, National Economic Survey |
| | Population of working adults (resident) | Persons | If applicable | Ministry of Manpower/Labour, National Economic Survey |
| | Average monthly wage | SG$/month | Calculated | Ministry of Manpower/Labour, National Economic Survey |
| | Average annual wage | SG$/year | Calculated | Ministry of Manpower/Labour, National Economic Survey |
| | Interest rate | % | Estimated | Literature review |
| | Discount rate | % | Estimated | Literature review |
| | Wage loss per death | SG$/death | Calculated based on average wage, average age, retirement age, discount rate | – |
| **Community testing** | Cost per swab test | SG$/test | Estimated | Hospital database, Ministry of Health |
| **Contact tracing** | Average monthly wage of contact tracers | SG$/month | Estimated | Ministry of Health, Ministry of Manpower |
| | Number of close contacts traced per case | Contacts/case | Estimated | Ministry of Health, European Centre For Disease Control and Prevention guidance on contact tracing |
| | Number of close contacts traced per officer | Contacts/officer | Estimated | Ministry of Health, European Centre For Disease Control and Prevention guidance on contact tracing |
| **Community isolation/ quarantine** | Number of close contact per positive case | Persons | Estimated | Ministry of Health, European Centre For Disease Control and Prevention guidance on contact tracing |
| | Numbers of quarantined days for close cases | Days | Estimated | Ministry of Health, European Centre For Disease Control and Prevention guidance on contact tracing |

Continued

**Table 2** Continued

| | | Units | Remarks | Sources |
|---|---|---|---|---|
| **Travel restriction** | Accommodation receipts loss | SG$/year | Estimated | External research databases, National Tourism survey, Economic surveys tourism sector |
| | Food and beverages receipt loss | SG$/year | Estimated | External research databases, National Tourism survey, Economic surveys tourism sector |
| | Shopping receipt loss | SG$/year | Estimated | External research databases, National Tourism survey, Economic surveys tourism sector |
| | Sightseeing, entertainment and gaming receipt loss | SG$/year | Estimated | External research databases, National Tourism survey, Economic surveys tourism sector |
| | Others receipt loss | SG$/year | Estimated | External research databases, National Tourism survey, Economic surveys tourism sector |
| | Monthly visitor | Visitor/month | Estimated | External research databases, National Tourism survey, Economic surveys tourism sector |
| | Average visitor's expenditure | SG$/visitor | Total tourism receipt/tourists | Based on tourism receipts and number of tourist arrivals: Total tourism receipt/number of tourist arrival |
| **Mask cost** | Cost per mask | SG$/item | Estimated | Marketplaces price, business to business platform prices with mark up for distribution (if applicable) |
| | Cost of masks distributed to the public per month | SG$/month | Calculated based on cost of mask, population and distribution mark-up factor | – |
| **Social distancing** | Number of social distancing officers | Person | Estimated | Ministry of Health |
| | Average monthly wage of social distancing officers | SG$/month | Estimated | Ministry of Health, Ministry of Manpower |
| **Cost due to as-needed school closure** | Probability of one case in school resulting in school cluster/school closure | Probability | Estimated | Ministry of Health, Ministry of Education |
| | Number of children per school | Children/school | Estimated | Ministry of Education |
| | Duration of school closure | Months | Estimated | Expert opinion |
| | Productivity loss of working dependents | % | Estimated | Literature review on productivity loss, human resource reports and business management studies |
| **Cost due to as-needed workplace closure** | Probability of one case in workplace resulting in workplace cluster/workplace closure | Probability | Estimated | Ministry of Health |
| | Number of people affected per closure | People/case | Estimated | Ministry of Health, Ministry of Manpower |
| | Duration of workplace closure | Months | Estimated | Ministry of Health |
| **Work-from-home (WFH) policy** | Productivity loss from WFH | % | Estimated | Literature review on productivity loss, human resource reports and business management studies |
| | Proportion of workers working from home | % | Estimated | Ministry of Health, Ministry of Manpower |
| **Impact on retails and services (limited social gathering)** | Loss in income from retail and other services | SG$/month | Estimated | National Economic Survey |
| | Loss in income from retail sales (under full circuit breaker measures) | SG$/month | Estimated | National Economic Survey |
| | Loss in income from food and beverage (under full circuit breaker measures) | SG$/month | Estimated | National Economic Survey |
| | Income loss after reopening | % | Estimated | National Economic Survey |

Continued

**Table 2** Continued

| | | Units | Remarks | Sources |
|---|---|---|---|---|
| **Economic loss from unemployment** | Total unemployment increase | Cases/month | Estimated | Based on National Economic Survey, reports, international databases |
| | Average duration of unemployment | Months | Estimated | Literature review based on unemployment and transitioning |
| | Total wage loss due to unemployment resulted over 12-month period | SG$ | Unemployment × wage × unemployment duration | Based on average wage and duration of unemployment Unemployment × wage × unemployment duration × time horizon |
| **Vaccines parameters** | Number of doses required | Doses | Assumed | WHO Target Product Profile, COVID-19 vaccine trials (when available) |
| | Efficacy | % | Estimated | WHO Target Product Profile, COVID-19 vaccine trials (when available) |
| | Proportion of population covered by vaccine | % | Estimated | Ministry of Health, WHO subnational vaccine coverage data |
| | Mild adverse reaction (injection reaction and systemic adverse reactions) | % | Estimated | WHO vaccine reaction rates, COVID-19 vaccine trials (when available) |
| | Severe anaphylaxis | % | Estimated | WHO vaccine reaction rates, COVID-19 vaccine trials (when available) |
| | Grade 3 adverse reactions requiring hospitalisation | % | Estimated | WHO vaccine reaction rates, COVID-19 vaccine trials (when available) |
| **Health-utility** | **COVID-19 related** | | | |
| | Healthy | QALY | Estimated | Literature review on COVID-19 health utilities |
| | Mild/moderate symptoms | QALY | Estimated | Literature review on COVID-19 health utilities |
| | Severe symptoms | QALY | Estimated | Literature review on COVID-19 health utilities |
| | Very severe symptoms | QALY | Estimated | Literature review on COVID-19 health utilities |
| | Death | QALY | Assumed as 0 | – |
| | **Medical isolation** | | | |
| | Disutility due to medical isolation | QALY | Estimated | Literature review on medical isolation |
| | **Vaccine-related side effects** | | | |
| | Mild adverse reaction (injection reaction and systemic adverse reactions) | QALY | Estimated | Literature review on influenza vaccination |
| | Severe anaphylaxis | QALY | Estimated | Literature review on influenza vaccination |
| | Grade 3 adverse reactions requiring hospitalisation | QALY | Estimated | Literature review on influenza vaccination |

QALY, quality-adjusted life years; SG$, Singapore dollars.

parameters and outcomes.[26] The study model will also be validated internally and externally: (1) face validity by clinical and modelling experts; (2) internal validation with checks for consistency, plausibility and debugging; and (3) external validation with published literature.[27]

### Sensitivity analyses

Sensitivity analyses is a critical component of a cost-effectiveness analysis to examine the large number of assumptions made as well as the uncertainties associated with model parameters. Threshold analysis may be performed to determine the threshold cost of the vaccine under the various target product profiles modelled. Scenario analysis will consider different time horizons, vaccine characteristics and vaccine prioritisation policies in this study.

Value of information analysis (VIA) could be applied to measure the gains from additional research which reduce the uncertainty in the model parameters. In particular, based on the results from sensitivity analysis, we could identify which parameters the ICER is most sensitive to. Those parameters could be considered for additional research to reduce uncertainties. Then we can decide whether to proceed with the additional research, by comparing the gains from the research (quantified in VIA) with the cost of conducting the research. If the additional research needed is prospective, VIA can also help to decide how much data to be further collected, thus optimising the research design.

One limitation of using VIA here is that due to the urgency of COVID-19, new policies may need to be implemented immediately with the emergence of new clinical

evidence. Hence there may not be sufficient time for additional prospective research before vaccine approval and distribution. However, VIA could help in at least two ways. First, if VIA shows that reducing the uncertainty for selective parameters is of high value, it may be necessary to review if all existing information of those variables have been used. Second, VIA could inform the optimal post-licensure data collection needs for monitoring the bene-fit–risk profile of vaccines, which includes any possible enhancement impact in post-licensure studies, and will help to determine future vaccination strategies beyond the initial doses.

## Patient and public involvement

Patients or the public will not be involved in the design, or conduct, or reporting, or dissemination plans of this research.

## DISCUSSION

Several methodological challenges exist for conducting an economic evaluation of COVID-19 pandemic response strategies. First, local data of the effectiveness of NPI may not be available for analysis. At the same time, effectiveness data of NPIs from another country setting are not directly applicable as they rely heavily on ecological factors, unlike pharmaceutical products that rely more on physiological and biological factors. Second, for vaccines, the long-term safety profile of COVID-19 vaccines will not be known, which may underestimate the risk associated with its use. In addition, uncertainties surrounding the long-term effectiveness of each vaccine candidate remains to be evaluated. Hence economic evaluation should be conducted once at the introduction of the vaccine with a subsequent analysis when more robust data is available. Third, the population group to receive priority vaccination in the setting of limited vaccine supply remains to be determined. Available vaccine prioritisation guidance by the WHO and Joint Committee on Vaccination and Immunisation have given health workers and the elderly the highest priority based on the goals of preserving health system capacity and minimising COVID-related deaths.[28 29] However, other jurisdictions may prioritise vaccination based on the goal of economic recovery, or a combination of both health-related and economic goals. These considerations would vary widely between jurisdictions, each with significant ethical considerations. Furthermore, other factors that may threaten vaccine uptake rates should be examined such as vaccine acceptance, payment models for vaccination and expectations of daily living post-vaccination. Yet, efforts for economic evaluations of these COVID-19 pandemic response strategies should not be hampered as extensive sensitivity and scenario analysis of these known variables that contribute to large uncertainties can be conducted. Hence, a greater challenge would then lie in the interpretation of the results considering all these limitations.

As the COVID-19 pandemic continues, there are many trade-offs that need to be made among the various policy options. Economic evaluation can provide one source of input to inform this complex decision-making process. The protocol that we have described provides a detailed guide to estimate the broader economic costs of various NPIs that are being used, in line with other publications that have recommended the need to consider broader societal costs.[30 31] Previous studies have so far established a cost inventory without specific guidelines or recommendations on how the costs should be estimated. Acemoglu *et al* presented optimal lockdown strategies in the USA using trade-off between mortality and economy protection efforts without explicitly deriving the costs and associated calculations from the relevant components.[32] Kim and Neumann proposed a cost inventory of policy responses for COVID-19 and highlighted the importance of the considering broader societal costs without guidance on how to compute the relevant costs.[33]

The costing framework that we have developed aims to fill such gaps and to provide a standard tool through which economic evaluations for COVID-19 vaccines can be based on. This framework also allows for the flexible analysis of different policy combinations according to a country's needs. Hence, the extent of relaxation of individual COVID-19 policies to be explored in the presence of a vaccine. By allocating specific cost associated with specific policies, this framework allows comparison between different policy combinations, but more importantly to compare the costs between different scenarios. This is key to policy evaluation where long-term planning is highly time-sensitive and policies should be allowed to have flexible implementation. In addition, this protocol would also be of relevant to settings with limited economic evaluation expertise, as data management plans can be drawn up to facilitate timely collection of these essential variables to help accelerate the decision-making process.

In conclusion, we have described a costing framework which will be useful to researchers and policymakers who are interested to evaluate the cost-effectiveness of various policy options. The costing framework is described in great details to allow researchers and policymakers to add on relevant cost parameters, which we have not considered, or remove cost parameters that are not relevant or are not available in their own jurisdictions.

**Contributors** HLW and HC conceived the study; HLW, HC and YT contributed to the design of the study and procurement of funding. BWBC, VAH, JL and FTG drafted the manuscript and all authors have read and approved the final manuscript.

**Funding** The study was supported by the National Medical Research Council, Ministry of Health, Singapore (COVID19RF3-0057). The National University of Singapore is a member of the International Decision Support Initiative, which facilitates funding support for publication fees from the Bill and Melinda Gates Foundation (OPP1202541).

**Disclaimer** The National Medical Research Council, Ministry of Heath, Singapore, and Bill and Melinda Gates Foundation have no direct role in the design of the study and collection, analysis and interpretation of data and in writing the manuscript.

**Competing interests** None declared.

**Patient consent for publication** Not required.

**Provenance and peer review** Not commissioned; externally peer reviewed.

**ORCID iDs**
Brandon Wen Bing Chua http://orcid.org/0000-0002-9559-4205
Hannah Clapham http://orcid.org/0000-0002-2531-161X

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
