## [Reviewer comments · BMJ Open]

ARTICLE DETAILS

TITLE (PROVISIONAL)	Protocol for the Economic Evaluation of COVID-19 Pandemic Response Policies
AUTHORS	Chua, Brandon; Huynh, Vinh Anh; Lou, Jing; Goh, Fang Ting; Clapham, Hannah; Teerawattananon, Yot; Wee, Hwee Lin

VERSION 1 – REVIEW

REVIEWER	Yaesoubi, Reza Yale University School of Public Health, Health Policy and Management
REVIEW RETURNED	11-Jun-2021

GENERAL COMMENTS	The authors describe a protocol for the economic evaluation of response policies for the COVID-19 pandemic in Singapore. I found the proposed study plan sound and reasonable and I believe this will be a valuable study. A few comments: 1. I would recommend authors to follow the CHEERS guideline in conducting and presenting the results of their economic evaluation analysis: https://www.equator-network.org/reporting-guidelines/cheers/.2. The authors are planning to evaluate the performance a large number of control policies including, community testing, contact tracing, isolation, travel restriction, mask use, social distancing, and several more, which in my opinion, would take a very long time to complete. I would recommend authors to limit the number of control policies to those that are still relevant at the time of the study.3. I would also recommend authors to use a formal calibration approach to estimate model parameters that are not observable or directly informed by data: https://pubmed.ncbi.nlm.nih.gov/28247184/.
--

VERSION 1 – AUTHOR RESPONSE

Response to Dr. Reza Yaesoubi, Yale University School of Public Health

1. I would recommend authors to follow the CHEERS guideline in conducting and presenting the results of their economic evaluation analysis:

It was indeed intended that we should follow the CHEERS guideline, as required by major clinical and health economics journals. We have updated the manuscript to reflect this change.

Page 4, Paragraph 5:

The methods and reporting of the study results will conform to the Consolidated Health Economic Evaluation Reporting Standards (CHEERS). Patients and the public will not be involved in the design, or conduct, or reporting, or dissemination plans of this research. The study will commence on 01 Oct 2021, with a proposed completion date of 30 Jun 2022.

2. The authors are planning to evaluate the performance a large number of control policies including, community testing, contact tracing, isolation, travel restriction, mask use, social distancing, and several more, which in my opinion, would take a very long time to complete. I would recommend authors to limit the number of control policies to those that are still relevant at the time of the study.

The study will report the most relevant control policies at the point of study completion.

The study methods have been updated to reflect this under the subsection "COVID-19 response policies (pharmaceutical and NPI) and its impacts". We have also revised Figure 1 to reflect the most relevant response policies for consideration.

Page 6, Paragraph 5:

The most relevant control policies at the time of study completion will be reported.

3. I would also recommend authors to use a formal calibration approach to estimate model parameters that are not observable or directly informed by data

Thank you for the suggested reference and comments. We have added a section on model calibration and validation (Page 15). Nonetheless, we are concerned if we will have the necessary data to do so.

VERSION 2 – REVIEW

REVIEWER	Yaesoubi, Reza Yale University School of Public Health, Health Policy and Management
REVIEW RETURNED	27-Aug-2021
GENERAL COMMENTS	The authors have addressed my comments.